# Social Interaction of Indonesian Rural Youths in the Internet Age

**Rista Ardy Priatama [1,\*]** , **Kenichiro Onitsuka [2,\*]** , **Ernan Rustiadi [3]** and **Satoshi Hoshino [2]**

1   Graduate School of Agriculture, Kyoto University, Kyoto 606-8502, Japan
2   Graduate School of Global Environmental Studies, Kyoto University, Kyoto 606-8501, Japan;
    shoshino@kais.kyoto-u.ac.jp
3   Faculty of Agriculture, IPB University, Bogor 16680, Indonesia; ernan@indo.net.id
*   Correspondence: rista.ardy.43n@kyoto-u.jp (R.A.P.); onitsuka@kais.kyoto-u.ac.jp (K.O.);
    Tel.: +62-853-1111-2256 (R.A.P.); +81-75-753-6158 (K.O.)

**Abstract:** The internet penetration on young villagers gives rise to question about its effects on the social interaction and behavior patterns as it accelerates the interaction with the wider network. However, the internet could possibly decrease both the social interaction with local people at the village and the dependency toward native villagers as internet utilization allows the users to be more selective in terms of interactions according to their interests. This research aims to examine the impacts of the Internet through a dystopian view by comparing the degrees of senses of place, participation in local activities, and social capital among internet and non-internet users using a statistical approach. The analysis of variance and linear regression were employed in the present study. The results revealed that the internet did strengthen both broad and local social capital. It also supported young villager's participation in local activities. Nevertheless, the internet was observed to decline the villager's sense of place, especially the desire to remain a resident in the native village. Better access to information and network gained by the users elevated their chances to move outside the village when better opportunities were observed elsewhere.

**Keywords:** internet use; young villagers; sense of place; local participation; social capital; rural life

## 1. Introduction

### 1.1. Background

Social life in rural areas on Java Island in Indonesia, in traditional view, is characterized by a high sense of closeness among residents because of pronounced intimate social interactions at the local community level [1]. Problems among villagers are resolved through traditional forums. Hinduism and Islam culturally influence their character of social life [1]. The latter is more dominant because it was introduced lately. The practices of religious, social gatherings routinely carried out and attended by residents are examples of the manifestations of the religions. Villagers are also aware of the term "gotong royong" or cooperation [2] representing activities performed together, generally in the form of physical activities (e.g. cleaning the environment, building village halls, and repairing other public facilities) without unduly expecting wages. Local activities and forums then historically strengthen the resilience of residents against common problems.

The presence of the internet in the countryside gives rise to discourse about whether rural social life is still relevant with the traditional view or not [3], especially on the younger generation that grew together with the development of internet and communication technologies (ICTs) [4]. Before the widespread use of telephones and the internet, rural residents managed their reciprocation through

face-to-face and traditional communication by which deep relationship could be considerably formed. This action strengthened social capital (SC) within local community [5] and reinforced one's attachment and dependency on the local community and living place. Later, the involvement of a resident on the internet afterwards allows him to be more selective in accordingly interacting with his interests [6]. It can accelerate, and shorten the time of, one's interaction with the network beyond space [7]. But, in the dystopian view, it possibly decreases the intensity of interaction with local people as a person needs time and effort as well to manage the other acquaintances [6].

Besides transformed social interaction, the immediate impact felt by internet users is the easy access to information. Abundant information can support young villagers in making decisions in daily matters [8–10]. The internet, with its various derivative applications, also modifies rural youth's ways to do activities such as shopping, playing games, reading, and watching videos by simply going online. Things brought about by the internet in some cases are the answers to the needs of villagers (e.g. Stern et al. [11]). For instance, the information on job vacancies or skill training can easily reach rural youths' gadgets. However, sources of prospective self-improvement carriers are generally outside the native village.

The new style of rural social life, however, could bring both negative and positive impacts. The internet inevitably improves people's networks, understanding of the outside world, lifestyle, and other personal aspects [7]. Since the rural areas mostly have limited facilities to meet a higher lifestyle, there would be a tendency in the people to feel more comfortable when interacting with resources outside the village. Consequently, villager's dependence on native villages becomes lower. Finally, the assumption to be examined in this research is that the internet on one side increases the autonomy of its users [3,7], but conversely decreases their social interaction with local residents and dependency on the place of origin. This issue at the local community level in rural areas are rarely discussed.

Reinforcement of social capital (SC) and participation in local activities of young villagers is useful for improving creativity in rural life. The sense of belonging of villagers to the current place is one of the keys to reducing migration outside the region and avoiding the leakage of potential human resources to the other areas. Managing these issues is challenging in the online age. The internet privileges young villagers to maintain their own interests and networks [12]. Yet it might weaken local ties, the existence of local activities and the sense of place at the village level. Therefore, there are three research questions to be addressed in the present study:

RQ1. What are the effects of the internet on villager's sense of place and participation in local events?
RQ2. Does the internet increase both broad SC and the local SC of the villagers?
RQ3. Are there specific purposes of the internet use affecting one's sense of place, local participation and SC?

Among current villagers, the younger residents refer to a group with the potential of experiencing significant internet impact because internet penetration in this group has reached three-quarters of the population under 35 years old [13]. This study observed the differences in patterns of social interaction and behavior between the internet and non-internet users among rural youth in local communities through the assessments of the senses of place, participation in local activities and SC. The latter was classified into local and broad SC. This research is useful in providing an overview of social life in rural areas in the coming decades because the Indonesian government is working to strengthen the internet network access in all regions through a national project, the so-called "Palapa Ring" [14]. Thus internet inevitably will be integrated into the life of villagers and gradually bring them into new socio-economic patterns [7].

*1.2. Research Framework*

Since the launch of the internet four decades ago, some people still haven't used it for various reasons, ranging from reluctance for the elderly to economic constraints. Therefore, people can be broadly grouped into internet users and non-internet users (see Figure 1). In urban areas, this division

among young people is no longer relevant, but it still occurs in the village youth group. Also, users according to their intensity in utilizing the internet, can be divided into heavy, moderate, and light users. The present study tried to observe the impact of the internet use by the villagers on their sense of place, local participation, local SC, and broad SC because these concepts are often regarded as the backbone of social life in rural areas. Local social capital here is SC gained by someone within village, while broad social capital is one's SC gained from acquaintances including friends and family living outside the village. This study tried to proof the dystopian view that the internet could bring undesired outcome i.e. a decline in the quality of rural social life.

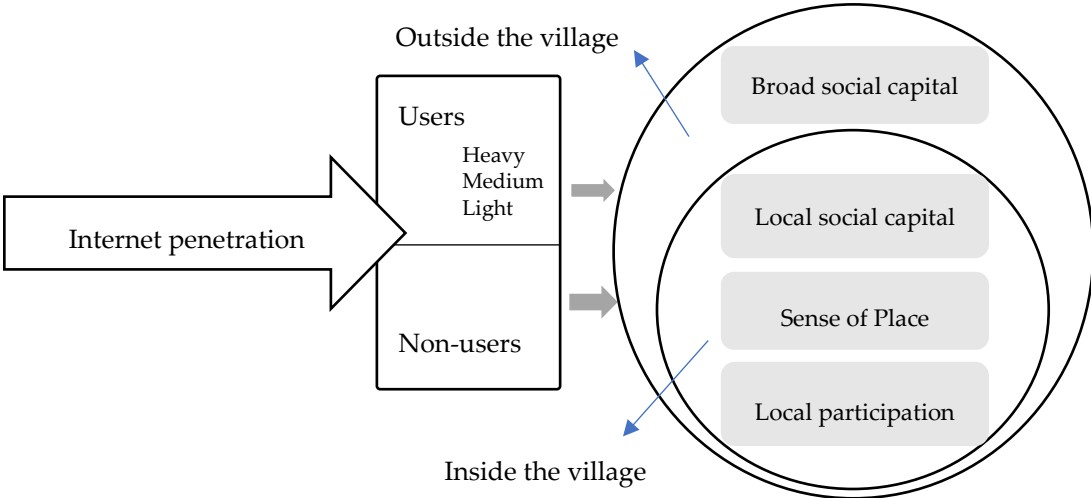

**Figure 1.** The framework of the research.

### 1.3. Literature Review

#### 1.3.1. Social Interaction in the Internet Era

Contemporary internet use is linked to networks of human relations. SC as a result of one's networked reciprocation is often used for describing the quality and quantity of social interaction. The increasingly competitive prices make the access to the internet not only limited to certain groups but also adopted by lower-class economic groups [15]. The benefits of the internet utilization are social connectivity, an increasingly dense network of interactions, and the strength of its bonds [16,17]. The involvement in the larger group is certain, and individual can choose the network that suits his interests [17,18]. Through social networking sites (SNS), one can communicate with friends and find friends of friends. Chat applications such as WhatsApp can now bring together people with an interest-based friend [19] whom he has never directly met.

The internet facilitates new forms of social interaction with existing ties, new ways to form bonds, and the possibility to strengthen old ties [20]. SNS [21–23] and chat applications [19] are the most influential internet platforms in shaping the social interaction as well as bonding and bridging of people today. In order to analyze people's relations in internet era, William [24] and Tiwari et al. [25] differentiated between online and offline relationship. Yet, Kennedy and Lynch [26] and Neves et al. [27] argue that distinguishing acquaintances based on online and offline acquaintances seems irrelevant since both have merged in the accumulation of interactions. Most people manage their offline ties via online interactions. Bridging without offline interaction is possible, but bonding without offline interaction is difficult to form.

For certain settings, the decline of relationship quality among people is possible to happen. Quan-Haase and Wellman [28] summarized the various impacts of internet usage in term of social interaction. The first is the internet transforming SC. The internet provides a means for convenient and inexpensive communication with far communities of shared interest. This leads to spatially dispersed

internet-based social networks, away from local or group-based solidarities. Second, the internet reduces the quality of social interaction through entertainment and information capabilities that attract the users from family and friends around. In individuals who experience this, meeting themselves with online leisure or interest groups facilitated by the internet decreases their involvement in the local neighborhood. Third, the internet adds or supplements SC because it has been fulfilled in life. It is used to maintain existing ties and civic engagement by utilizing electronic contact features. In a certain relationship environment, it is used by individuals to arrange regular meetings and to channel their hobbies.

### 1.3.2. Rural Life in the Internet Era

The definition of rural areas based on their distance from cities is now often questioned after the spread of communication technology networks [12]. The internet accelerates the transport and exchange of sound or video at speeds that make distance irrelevant [29]. Online communication can be used flexibly and comfortably as access can be done at wish. Unlike television and telephone, the internet presents various directions of communication: one to one, one to many, many to one, and many to many. Consequently, it disrupts the traditional view regarding rural area. While the internet directs villagers to produce new structures of society, it creates rural areas with new social, economic, and cultural challenges [12].

Slowly but surely, along with the increase in internet users, rural areas are in transition in which a lot of rural activities are fused with the internet [3], thus, creating a dependency on technologies [12]. Even so, some groups in rural areas still seem to be slow in welcoming the benefits of the internet, including adults to old age, residents with poor economic conditions, and people with low literation [29]. Over time, the lack of access to the internet tends to increase one's loss. If someone needs ICT to get an education, a job, order an appointment for government services, or support the sale of a small company, then not having access to the internet can be an obstacle to productive involvement in the world. As people increasingly use the internet to share information, the inability to access such information can create barriers to public participation [12].

Even though the internet can present a lot of potentials, its economic impact for rural communities still requires time. Some e-platforms, for example, have tried to help farmers on marketing their fresh agricultural products. But, there are still large gaps between farmers and platform providers where farmers have not been able to utilize these services due to limited digital literacy and ability [30]. However, in the online era, barriers to distance from markets can be reduced in building an entrepreneurial culture. People can take advantage of online trading networks to increase income by being a re-seller or selling products on e-marketplaces.

### 1.3.3. Sense of Place

One's residence and interactions basically influence the sense of place. A person can have a personal attachment to a limited geographical location as a result of the activities carried out at such a place [31]. Periodic or daily physical contact with a place is the key to maintaining the senses of the place as well as other relationships [31]. This suggests that the degree of sense of place varies among the population based on quantity of acquaintances they assort to as well as interaction range and intensity. Place identity and place dependency are known as components of the sense of places [31–33]. The first refers to the dimensions of the self, such as the mixing of feelings about the physical state and the symbolic connection to the place that defines who he is [34], while the latter describes an occupant's perceived strength of association between him- or her-self and specific places [35]. Place dependence refers to a strong sense of association with the place concerning a goal-oriented and resource-oriented matters in a place that can support [36].

The dimension of dependence on the place is descriptive based on individual, aggregate, or group relations. There are two components used to assess a person's dependence: the quality of the current place and the relative comparison on the quality of the current to alternative places [35].

In evaluating the quality of places, individuals assess how well they can actualize their goals and activities. The results of this assessment determine someone's satisfaction with the place. Such level of satisfaction is indexed by the extent to which the quality of the place deviates from the level of occupant comparison for the place [35]. The better a place can realize one's interests, the greater one's satisfaction will be. However, one's judgment on a place is dynamic. Good and bad experiences, interests, beliefs, potential uses, and knowledge of alternatives are all factors affecting this dynamic.

### 1.3.4. Rural Traditional Events

Discussion and literature on local participation in rural areas mostly discuss the involvement of villagers in political and economic activities. From an economic perspective, participation was initially emphasized in the capitalization of rural resources to support the downstream industry. But lately, rural development perspectives have been directed to autonomy by applying various economic concepts such as agropolitan, ecotourism and cultural tourism [37]. Whereas the political perspective emphasizes the activeness of the community in political or ideological activities. The discussion of local participation related to purely cultural or voluntary activities without economic and political frills is still very rarely discussed [11].

Indonesia's rural areas have their own tradition of social culture endowed by an elder generation, that might be disrupted in internet era. There are certain events oriented towards social gatherings wrapping up with specific themes such as religion or leisure. The attendance of villagers at those events correlates with the closeness and social trust among the residents. Active participation of villagers in joint social activities can reduce social problems arising from economy, politics, and environment [38]. The low level of community participation in cultural activities has led to the loosening of ties between them, having an impact on decreasing social cohesion and the sense of identity [39].

Community involvement is the action of people to actively and collectively participate in activities or programs [39]. The intensity of the involvement of everyone varies, from people who passively benefit to those who are active in making decisions. The existence of social interaction distinguishes between "people" and "community" because society only appears when there are social ties between residents [40]. Non-profit activities in rural areas are instruments for maintaining social ties and manifestations of mutual assistance activities among residents in the case of death, marriages, fire, stealing problems, and others.

### 1.3.5. Social Capital

Social capital is formed due to interaction, committed norms, and trust toward other individuals inside the circle, while the density of networks determines its accumulation. In the individual sphere, one interacts with others in places such as school, neighborhood, organization, etc. Because of the sense of good will humans have, each follows and applies the applicable norms—e.g., help, support, sympathy, greeting, and not to cheat—when interacting with the others in the circle in which they are involved. This aspect implies that there exists reciprocity of norm-related actions from which trustworthiness comes. However, trust degree obtained by an individual from others differs depending on their efforts or investment in increasing interaction, applying the norms, and widening the networks.

The intensity differences in forming SC produce a variety of social ties. Strong reciprocity and trustworthiness among individuals create exclusive intercourse, the so-called "bonding," whereas inclusive relationship, commonly mentioned as "bridging", is the result of thin reciprocity and credence [16,41]. Bonding provides emotional or instrumental support from one's exclusive tie [24,27] such as family or friends. It is a strong-tie [42] and good for undergirding reciprocity and mobilizing solidarity [16]. By contrast, bridging is good for connection to external assets that link somebody to distant acquaintances or move him in a new different circle [16]. Most people get job information as one of the desired outcomes from this weak tie [43]. For short, both strong and weak ties of SC have their crucial function for individuals.

That values that exist within a social network is the core idea of the SC theory [16]. It refers to tangible substances in network ties: goodwill [44], mutual support, shared language, shared norms, social trust, and sense of mutual obligation from which people can obtain value [45]. The basic definition of SC in the individual sphere is provided by Bourdieu [46]: "aggregate of actual and potential resources which are linked to possession of durable network or more or less institutionalized relationship of mutual acquaintance and recognition." He stresses the term "resources" as a result of SC, which is achievable if somebody engages himself for being a member of certain groups or circles.

## 2. Materials and Methods

### 2.1. Study Area

This study was done in one of the villages in Probolinggo Regency, East Java Province, Indonesia. This Regency consists of 24 districts covering 330 villages in total. The government divides those villages into urban and rural areas, 95 and 235, respectively [47]. The agricultural sector remains eminent toward others, supporting 35.9% of the GRDP [48]. Thus provincial government designates this regency as the Agropolitan area in the regional spatial plan of East Java Province [49]. Most of the people (41.8%) were also working in this sector [48]. Hence, Probolinggo Regency represents the typical local region on Java Island, that contains many rural circumstances, in which agriculture plays an important role.

Tamansari village, precisely the part of Kraksaan District, was chosen as a specific area of the present study. The selection of villages was carried out by considering the prosperity level. In Tamansari Village, the number of lower-middle families, i.e. the sum of pre-prosperous families and prosperous families 1 (see BKKBN [50]), contributed to half of the total families (55.8%) based on the data from the local Government's Statistics Agency [51]. This village is also neither a district town nor a full-facility area. Regarding internet facilities, there was not any internet café there and free internet could only be accessed from one coffee stall. Young villagers generally accessed the internet through their smartphones. However, the strength of the broadband signal was fit for use from sending simple chat to watching videos on Youtube. Different from the older generation, young villagers having done their school mostly did not work in on-farm agriculture due to the lack of access to arable land and insufficient income. They in most cases commuted almost every day or were still looking for a job.

### 2.2. Data Collection

This study uses statistical approaches and only focuses on rural youth aged 15 to 34 years because this is a productive age range with a high level of internet penetration today. This range group also represents the face of the future of rural Indonesia. Semi-structured interviews with 4 key informants were conducted to obtain a picture of the mobility of the sample candidates. They included two village government employees and two neighborhood leaders. A preliminary survey with 10 people was also conducted to estimate the duration of filling out the questionnaire and to ensure that respondents could fill in comfortably. The actual population aged 15 to 34 years old was 312 people in February 2019. Because data collection was only limited to a month within a village, from the end of January to the end of February 2019, the authors targeted a sample of at least 50 percent of the population. The three people assisted the author in the survey encompassing two people with bachelor's degrees and 1 village employee.

Samples were randomly selected based on a list obtained from village officials. Each person was coded and then raffled. One hundred fifty-six prospective samples were first selected. Because this figure could be achieved and time was still available, the authors added several more respondents to be surveyed. Finally, it was managed to survey a total of 192 respondents, but five of them did not provide complete information. Therefore, there were 187 samples included in the analysis covering 139 internet users (74.3%) and 48 non-internet users (25.7%). All study participants provided informed consent.

### 2.3. Variables and Measurements

2.3.1. Demography

As a basic dimension, the respondent's demography is listed and presented. The demographic and inherent socio-economic background of people become notable factors when addressing ICT [52] and can be used to control the variables [53] to get more accurate results. In the questionnaire, these variables were asked: gender, age, education level, marital status, involvement as village organizer, and monthly spending (in IDR). The 'village organizer' is a question about whether a respondent holds a position or not in any village's formal or informal organizations. This question was prepared to control the people's participation as one active in village organization is more likely to attend the village's events. Assessment of the economic level of people uses 'monthly spending' instead 'personal income' since the samples predictably would contain a considerable portion of household and jobless without income. Another reason was that assessing personal income was a sensitive thing in certain regions in Indonesia.

2.3.2. Internet Usage

By considering its impact on social interaction, the purpose of using the internet was divided into 6 aspects: communication, entertainment, games, self-expression, self-improvement, and self-earning. Communication is the basis for the use of ICTs. The use of the internet for communication emphasizes the delivery of one person to another party via the internet. So, there are interactions between individuals and other individuals or individuals and groups. Entertainment purpose is the use of the internet for any form that consciously or unconsciously aims to entertain a person, ranging from simply filling spare time by viewing feeds on Instagram to watching videos on the service provider of films or YouTube. The game was separated from the entertainment group because dozens of studies showed that it has certain effects on a person's social behavior (see Uz and Cagiltay [54]). Self-expression is any form of delivery of thoughts and expressions through the internet. The practices include updating status, releasing posts on blogs or SNS, and commenting on other people's writings, works or stories. Meanwhile, self-improvement relates to the use of the internet to develop either soft skills or hard skills. Lastly, self-earning is the use of the internet to make money, starting from blogging involving AdSense to selling on e-marketplaces.

The interval scale was used to measure the intensity of using the internet for those purposes. This scale includes 7 ordinal levels ranging from never (1) to several times a day (7). So, the question about internet use was, "how often do you rate your internet usage for these purposes?" (see File S1, supplementary materials). In addition to the intensity of the usage, the scope of one's communication is also asked to estimate the network of acquaintances of someone who is actively communicating via the internet. The question was, "If you have, how do you rate the number of acquaintances whom you have routine contact with via the internet in each of the following categories?". 10 categories were mentioned (circles) and were to be filled by the respondent. These circles follow our SC measurements. For this question, 6 ordinal scales were used starting from 0 for none till 5 for a lot.

*Intensity*: How do you rate your internet usage for the following purposes?

(1) Entertainment (e.g., watching online video, surfing on SNS, reading novel, etc.)
(2) Playing games
(3) Posting new status or article (e.g., in Instagram, blog, Whatsapp, Facebook, etc.)
(4) Communication with friends and acquaintances
(5) Self-development (e.g., online course, searching for training, reading books, etc.)
(6) Online selling or business development

*Communication range*: If you have, how do you rate the number of acquaintances whom you have routine contact with via the internet in each of the following categories?

### 2.3.3. Sense of Place

The sense of place in this study refers to the initial concept of place, especially the one related to the concept of place dependency without including place identity. The importance of this was adjusted to research seeking to show the impact of the internet on a person's dependence toward the place of origin. The internet can connect young villagers with the expected interests and resources by reducing the distance barriers. They having successfully exploited the internet to mine these advantages would have a greater dependence on the outside world. This sense of dependency competes with that of the native village because in general, villages' condition is fainter in meeting youngster needs such as qualified education, job vacancy and lifestyle.

The place dependency in this study included only 3 components covering satisfaction on economic welfare, satisfaction on facilities, and desire to stay in the village. The author only included 2 questions in each component to avoid repetition. Satisfaction on economic well-being assesses the respondent's satisfaction in terms of income and adequacy in fulfilling lifestyle. Satisfaction in facilities is explored by asking respondents' judgment on the current facilities around the village. Then, the desire to remain in the village is accessed through questions about the degree of potential for moving to another place and confidence to get comparable happiness when taking decision to reside outside the village of origin. The following are questions for the sense of place with the 5-Likert-scale answer (see File S1):

*Facility satisfaction* [55]

(1)  Facilities and services of education, health, and infrastructure in my living place are good enough
(2)  Facilities such as shop matters, arts, and sports around my living place could meet my needs.

*Job Satisfaction*

(1)  My (family) income now is enough to meet my personal needs and lifestyle [55]
(2)  I can find a job that suits my preference in the area where I live

*Staying intention* (reversed) [56]

(1)  I am willing to leave my current residence if I find a comparable job or place in another area
(2)  I will remain happy if I must reside far from my current residence

### 2.3.4. Local Participation

A list of routine village activities was accessed through interviews with key informants. Tamansari Village residents still had traditional activities for internal villagers. The existence of collective activities here, except for economic or political reasons, is motivated by religious traditions, leisure time, environmental problems, and social innovation. The routine implementation of religious rituals in Tamansari is influenced by the culture of the Nahdiyin, the largest Islamic mass organization in Indonesia. Social activities leading to solving environmental problems and social innovation are driven by formal organizations overseen by the village government. Then, in leisure activities, villagers like to spend time together mostly through sports.

Current shared activities in Tamansari Villages can be divided into 2 according to schedule, namely weekly and annual events. The weekly program is in the form of dhikr, recitation of the Holy Qur'an, and sports. Men and women have different schedules to perform these activities. Meanwhile, the annual activities include neighboring meetings, village youth meetings, commemoration of the republic's independence, helping the festivity of the people, cleaning up the village, and Eid al-Adha celebration without separation between men and women in implementation. The questions and list of local events that respondent must use to define their participation (ordinal-scale answer from 1 for never to 5 for always) are as follows (see also File S1):

*Weekly events: In the last 3 months, how do you rate your presence in the following events in your village?*

(1)　　Prayer recitation
(2)　　Quran recitation
(3)　　Exercise

*Yearly events: In the last 2 years, how do you rate your presence in the following events in your village?*

(1)　　Neighborhood meeting
(2)　　Youth meeting
(3)　　Independence Day celebration
(4)　　Helping in neighbor's festivity
(5)　　Cleaning village
(6)　　Eid Al-Adha event (slaughtering sacrificial animals)

2.3.5. Social Capital

This study used three dimensions of SC: bonding, bridging, and social support. Bonding deals with resources potentially available in strong ties (such as family members, close friends, and close colleagues) and can be mobilized when needed [27]. It is the main provider of social support, ranging from emotional to financial assistance. Bridging deals with resources potentially available in weak ties such as acquaintances [27]. Because weak ties are generally heterogeneous rather than strong bonds, they have access to more varied resources such as information for work matters and wider social perspectives. Meanwhile, social support is a concrete item used to ask the closest people who can or are willing to provide assistance if respondent asks [57].

The structure of the SC assessment follows the measurement principle established by Chen et al. [58] with modification. They distinguished the acquaintance owned by someone into circles, family members, relatives, people in neighborhood, friends, coworkers, and old friends. In measuring our SC, bonding, bridging, and social support are differentiated based on the degree of relationship between individual and other individuals by following the concept of Granovetter's [43] weak ties and strong ties. So, in a relationship circle, someone can have acquaintances that can be grouped into bonding, bridging, and social support. Therefore, one's knowledge about and involvement in the organization are no longer considered as bridging (like the work of Chen et al. [58]), but rather deemed as a circle in which one obtain either strong or weak ties.

In the Indonesian context, the circle from which one gets social relations can be divided into 10, namely local communities, school-related circle, work-related circle, extended family, involvement in inter-regional organizations (cultural or political), supporting activities, online communities, college-related circle, previous neighbors and other acquaintances. Everyone has several engagements in different circles. A working person has a greater quantity of acquaintances compared to homemakers. So, working people had advantages due to the number of acquaintances. Using this approach, a distinction between a person's SC with surrounding villagers and that with an outside environment could be reached in order to answer the second research question.

Bridging in this study consisted of 2 constituent variables, i.e. the number of acquaintances and general trust. The number of acquaintances asks how the respondent rate the number of acquaintances he has in each circle, while the general trust asks the number of acquaintances trusted as good people in each circle. Bonding consists of close friends and acquaintances who often meet each other. The questions of social support cover financial, skill, mental, and energy support [57]. All questions for SC assessment (with a 6-ordinal scale answer from 0 to 5) were as follows:

*Bridging* [58]

(1)　　If you have, how do you rate the number of acquaintances in each of the following categories?
(2)　　Among the people in each of the following categories, how many can you trust?

*Bonding* [58]

(1)　Among the people in each of the following categories, how many are your close friends?

(2)　With how many people in each of the following categories do you keep a routine contact?

*Social Support* [57]

(1)　How many people in each circle will give you a financial loan in case you request it?

(2)　How many people in each circle will give you personal skill support in case you request it?

(3)　How many people in each circle will give you mental support in case you request it?

(4)　How many people in each circle will give you energy support in case you request it?

*2.4. Analysis*

The resulting approach used in this study was quantitative by involving several statistical methods. To answer the first research question (RQ1), the accumulation score of sense of place and local participation between internet users and non-internet users was compared using the analysis of variance. This statistical method was also used in describing the comparison of the strength of local and broad SC between internet users and non-internet users. To answer the second question (RQ2), the relevant circles to equalize the source of SC for all respondents were determined. Finally, for the third question (RQ3), the internet use (independent variable), including usage intensity and communication range, was regressed with a sense of place, local participation, local SC, and broad SC (dependent variable) using stepwise multiple regression. All statistical analyses were performed in SPSS 22 (IBM Corp., Armonk, NY, USA).

## 3. Results

*3.1. Samples*

3.1.1. Respondents' Characteristics

The respondents in this study were 187 people consisting of 38.5% female and 62.5% male. 56.1% of them were married (see Table 1). The age ranges covering respondents were divided into 5 groups: 15–18, 19–22, 23–26, 27–30, and 31–34 years old. The first group represented teenagers who were generally still in high school, while the second group consisted of respondents who were currently active as college students, working people, or jobless. The highest portion of respondents, based on education, was the group of senior high-school graduates (36.9%). However, elementary school (25.1%) and junior-high-school (26.7%) graduates remain high. Since there were several formal and informal institutions or organizations in Tamansari Village, some respondents (8.6%) were also engaged as members. 35.3% of respondents had monthly expenditures ranging from 500k to 1000k IDR, and 31.0% claimed to spend money only under 500k IDR.

The internet penetration of respondents, portion of internet users towards total samples, was 74.3%. Such a rate is still similar to those of researches conducted in Indonesia a few years ago (see APJII [13] and Onitsuka et al. [52]). According to gender, internet penetration in males (80.0%) was higher than that of female (65.3%). Household or housewife group contributed to a considerable portion of the low internet use rate of women. 44.4% of women were housewifery, and 56.3% of them were not internet users. Therefore, in this research, there was a correlation between gender and rate of internet usage ($r(187) = 0.164$, sig. (2-tailed) $< 0.01$, 0 for female, 1 for male).

Education and age still become key factors determining gaps in internet usage (see also Neves et al. [27] and Onitsuka et al. [52]). People with a younger age had a higher rate of internet penetration compared to older people ($r(187) = -0.443$, sig. (2-tailed) $< 0.01$). In the group of 15−18 years−old villagers, almost all members (96.8%) were users. Such a rate decreased in the older group and reached less than a half in the 31−to−34−years−old group (46.5%). Conversely, education level had positive impact on the possibility to use the internet ($r(187) = 0.532$, sig. (2-tailed) $< 0.01$). The portion of non-internet users decreased as education level gets better, from elementary school, junior-high-school,

and senior-high-school graduates, 63.8%, 28.0%, and 5.8%, respectively. Respondents who graduated or currently being at the university were all internet users.

**Table 1.** Demography of respondents based on users and non-users of the internet.

| Demography | Non-User | | User | | Total | | Demography | Non-User | | User | | Total | |
|---|---|---|---|---|---|---|---|---|---|---|---|---|---|
| | **n** | **% [a]** | **n** | **% [a]** | **n** | **% [b]** | | **n** | **% [a]** | **n** | **% [a]** | **n** | **% [b]** |
| *Total samples* | 48 | 25.7 | 139 | 74.3 | 187 | 100.0 | *Village organizer* | | | | | | |
| *Gender* | | | | | | | Member | 6 | 37.5 | 10 | 62.5 | 16 | 8.6 |
| Female | 25 | 34.7 | 47 | 65.3 | 72 | 38.5 | Nonmember | 42 | 24.6 | 129 | 75.4 | 171 | 91.4 |
| Male | 23 | 20.0 | 92 | 80.0 | 115 | 61.5 | *Occupation* | | | | | | |
| *Age (years old)* | | | | | | | Student | 1 | 3.4 | 28 | 96.6 | 29 | 15.5 |
| 15–18 | 1 | 3.2 | 30 | 96.8 | 31 | 16.6 | College student | – | – | 9 | 100.0 | 9 | 4.8 |
| 19–22 | 6 | 11.8 | 45 | 88.2 | 51 | 27.3 | Private employee | 2 | 8.3 | 22 | 91.7 | 24 | 12.8 |
| 23–26 | 4 | 16.0 | 21 | 84.0 | 25 | 13.4 | Farmer | 20 | 60.6 | 13 | 39.4 | 33 | 17.6 |
| 27–30 | 14 | 37.8 | 23 | 62.2 | 37 | 19.8 | Entrepreneur | 5 | 14.7 | 29 | 85.3 | 34 | 18.2 |
| 31–34 | 23 | 53.5 | 20 | 46.5 | 43 | 23.0 | Public employee | – | – | 4 | 100.0 | 4 | 2.1 |
| *Education* | | | | | | | Temporary worker | 1 | 10.0 | 9 | 90.0 | 10 | 5.3 |
| Elementary school | 30 | 63.8 | 17 | 36.2 | 47 | 25.1 | Jobless | 1 | 8.3 | 11 | 91.7 | 12 | 6.4 |
| Junior high school | 14 | 28.0 | 36 | 72.0 | 50 | 26.7 | Household | 18 | 56.3 | 14 | 43.8 | 32 | 17.1 |
| Senior high school | 4 | 5.8 | 65 | 94.2 | 69 | 36.9 | *Monthly spending* | | | | | | |
| Bachelor & above | – | – | 21 | 100.0 | 21 | 11.2 | <500k | 11 | 19.0 | 47 | 81.0 | 58 | 31.0 |
| *Marital status* | | | | | | | 500k–1000k | 21 | 31.8 | 45 | 68.2 | 66 | 35.3 |
| Single | 4 | 4.9 | 78 | 95.1 | 82 | 43.9 | 1000k–1500k | 9 | 24.3 | 28 | 75.7 | 37 | 19.8 |
| Married | 44 | 41.9 | 61 | 58.1 | 105 | 56.1 | 1500k–2000k | 5 | 35.7 | 9 | 64.3 | 14 | 7.5 |
| | | | | | | | 2000k–25000k | 2 | 20.0 | 8 | 80.0 | 10 | 5.3 |
| | | | | | | | 2500k–3000k | – | – | 2 | 100.0 | 2 | 1.1 |

[a] Percentage toward the total of each category ('total' column). [b] Percentage toward total sample number (187).

Internet penetration varied according to the types of work. All respondents who were active as public employees and students at the university were internet users. Nearly all students (96.6%) have dealt with the internet. The group of private-sector employees, entrepreneurs, temporal workers, and unemployed were at the same level of internet usage (around 90%). Between all occupation groups, the lowest penetration level was experienced by farmers (39.4%) and households (43.8%). Hence, farmers and households contributed to a large portion of the number of non-internet users.

### 3.1.2. Internet Use by Young Villagers

Communication and entertainment were the main purposes of people using the internet. 69.8% of the users communicated through the internet more than once a day and the rests mostly ranged from once per day to once a month (see Table 2). Young people, as shown by another survey [59], presume communication through the internet as an important behavior for their social life and they use it to shape and maintain social relationships. Then, users who entertain themselves through content on the internet at least once a day reached 87.1%. Age is the main indicative factor in this usage (r(139) = −0.218, sig. (2-tailed) < 0.01). This denotes that younger people spent a longer time entertaining themselves via the internet than the older ones.

**Table 2.** Respondents' intensity in accessing the internet for specific purposes.

| Purposes of Internet Use | User Number * Based on Intensity (Percent) | | | | | | |
|---|---|---|---|---|---|---|---|
| | **Never** | **Annual** | **Every 6 Months** | **Monthly** | **Weekly** | **Once per day** | **Several Times per day** |
| Communication | 2.2 | 0.7 | – | 2.9 | 12.2 | 12.2 | 69.8 |
| Entertainment | 1.4 | – | 0.7 | 3.6 | 7.2 | 21.6 | 65.5 |
| Game | 46.8 | 1.4 | 0.7 | 3.6 | 10.1 | 12.2 | 25.2 |
| Self-expression | 15.1 | 1.4 | – | 4.3 | 28.1 | 22.3 | 28.8 |
| Personal development | 53.2 | 0.7 | 2.2 | 7.9 | 9.4 | 13.7 | 12.9 |
| Self-earning | 66.9 | 3.6 | – | 5.0 | 5.0 | 2.9 | 16.5 |

* Internet user only: 139 respondents.

Internet utilization related to self-expressions was the next frequently used by internet users with lower intensity compared to the purpose of communication and entertainment. Only 28.8% of

respondents did this kind of usage several times a day, while 22.3% and 28.1% shared their thoughts, feelings, or moments, weekly and daily, respectively. The younger group tended to express themselves more often than the older group (r(139) = −0.204, sig. (2-tailed) < 0.05). While education level played a role with positive expression-related usage (r(139) = 0.351, sig. (2-tailed) < 0.01), but the educational gap between the younger and older group had contribution in this situation.

Even though playing games on the internet-based platform was popular, almost a half of internet users (46.8%) never did that. However, there were still a remarkable number of gamers with varying intensity from several times per day (25.2%), once per day (12.2%), to weekly (10.1%). Male played games through the internet more often than female (r(139) = 0.270, sig. (2-tailed) < 0.01, 0 for female, 1 for male). Playing game also had negative correlation with age (r(139) = -0.204, sig. (2-tailed) < 0.01).

The internet uses for self-improvement and self-learning are kinds of advanced utilization since the users necessarily need certain skills to grab this broad internet potential. Such kind jobs had positive relation to education level (r(139) = 0.388 for self-improvement, r(139) = 0.257 for self-earning, both sig. (2-tailed) < 0.01). Better education affected the cultural capital of individuals to develop their creativity in managing the available resources [46], especially the internet potential in this case. Young villagers seemed to have a lower rate of accessing the internet for those purposes. 53.2% of respondents never use the internet for skill- or knowledge-upgrading efforts. Only 12.9%, 13.7%, and 9.4% of the users improved themselves through the internet respectively in more-than-once-a-day, daily, and weekly basis. Even for self-learning purposes, villagers who never used the internet for earning money reached 66.9%. Only around one-third of samples (29.4%) did this occasion, ranging from once-a-day to monthly basis.

*3.2. Internet Impact on Sense of Place & Participation (RQ1)*

3.2.1. Internet Impact on Sense of Place

In general, there are considerable differences in the pattern of sense of place based on variables listed between internet users and non-users. Most users and non-users agreed that the facilities around their living area were good enough (see Table 3). The non-users who agreed with the claim of good quality of the facilities related to education, health, and other infrastructures (facility satisfaction 1) were 70.8%. There was not the non-users marking disagreeing and mildly agreeing in this question, but some, 6.3%, claimed that they mildly disagree. Then, 79.2% and 10.4% of non-users stated, respectively, that they were satisfied and mildly satisfied with the facilities related to lifestyle and socioeconomic needs such as shopping matters, masjid, and sport or art facilities (facility satisfaction 2). Even though the users had high agreement on the goodness of facility (both facility satisfaction 1 and 2) around them, the portion was lower than that of the non-users. Regarding facility satisfaction 1, 56.8% said to agree, and 6.5 % said mildly agree. Meanwhile, in facility satisfaction 2, respondents saying agree and mildly agree were 65.5% and 7.2%, respectively. Some users stated disagreement on the goodness of the facility quality around them: 5.8% disagree and 10.1% mildly disagree for facility satisfaction 1, 2.2 %disagree, and 8.6% mildly disagree for facility satisfaction 2.

**Table 3.** Percentage of respondents' answers to questions regarding place attachment.

| Sub-Variables & Number of Questions | User (%) | | | | | Non-User (%) | | | | |
|---|---|---|---|---|---|---|---|---|---|---|
| | Disagree | Mildly Disagree | Neutral | Mildly Agree | Agree | Disagree | Mildly Disagree | Neutral | Mildly Agree | Agree |
| Facility satisfaction 1 | 5.8 | 10.1 | 20.9 | 6.5 | 56.8 | – | 6.3 | 22.9 | – | 70.8 |
| Facility satisfaction 2 | 2.2 | 8.6 | 16.4 | 7.2 | 65.5 | – | 6.3 | 4.2 | 10.4 | 79.2 |
| Job satisfaction 1 | 5.8 | 3.6 | 36.7 | 12.9 | 41.0 | 2.1 | 8.3 | 52.1 | 2.1 | 35.4 |
| Job satisfaction 2 | 13.7 | 16.5 | 25.2 | 7.2 | 37.4 | 4.2 | 21.9 | 41.7 | 4.2 | 27.1 |
| Staying intention 1 | 29.5 | 9.4 | 19.4 | 21.6 | 20.1 | 2.1 | 2.1 | 16.7 | 31.3 | 47.9 |
| Staying intention 2 | 25.2 | 10.1 | 27.3 | 19.4 | 18.0 | 2.1 | 2.1 | 10.4 | 27.1 | 58.3 |

For non-internet users, job satisfaction, consisting of income sufficiency to meet lifestyle (job satisfaction 1) and compatibility of the current job to the desired job (job satisfaction 2), become the

least agreed factors. 35.4% and 2.1% of respondents claimed agree and mildly agree for job satisfaction 1, while 27.1% and 4.2% stated agree and mildly agree for job satisfaction 2. Most non-users (52.1%) felt neutral about the question regarding income adequacy. However, considerable disagreement of nonusers appeared in the compatibility of the current job, 22.9% and 4.2% claimed to disagree and mildly disagree, respectively. The users tended to have higher job satisfaction in both job satisfaction 1 (41.0% agree, and 12.9% mildly agree) and job satisfaction 2 (37.4% agree, and 7.2% mildly agree). Nevertheless, the disagreement on the asked statements was also higher, reaching in total (disagree plus mildly disagree) 9.4% for job satisfaction 1 and 30.2% for job satisfaction 2.

Regarding the degree of staying intention, there is a clear gap between internet users and non-users. This dimension asked the respondent regarding the possibility to reside in the village even though they had better life choices outside (staying intention 1) and unhappiness when he or she must move out of the village (staying intention 2). The non-users had a higher possibility to remain in the village as four-fifth of the respondents showed agreement with this statement, precisely 47.9% agreed, and 31.3% mildly agreed. The unhappiness degree to move outside the village was also high, i.e., 58.3% claimed to agree, and 27.1 % disagree. Only a little portion expressed disagreement on this issue, 2.1% in both disagreeing and mildly disagreeing statements on staying intention 1 and staying intention 2. Conversely, only two-fifths of the users would remain in the village (21.6% agree, and 20.1% mildly agree), and almost two-fifths of the users had a high possibility of moving out of the village (19.5% agreed, 9.4% mildly disagreed). Even 35.3% believed that they would also get happiness while residing outside the place of origin.

The variance analysis was performed to examine the gap of sense of place between internet users and non-users. Based on this test (see Table 4), there was an obvious difference between facility satisfaction (significant at 0.05 level) and staying intention (significant at 0.01 level). Even though there was a clear difference in the pattern of job satisfaction based on agreement percentage, this variable did not differ among users and nonusers. In facility satisfaction, the user group had a lower mean (8.24) score compared to that of the nonusers' (5.88). The mean score of the user group (5.88) was also lower than that of the non-user group (8.58) on the degree of staying intention. But it is necessary to consider the bias potential since there was some relationship found in the correlation test between a sense of place and demographic variables. Facility satisfaction positively correlated to gender and negatively to education. Meanwhile, staying intention had a relation to gender, education, age, and membership in village organization. The first two were negative, and the other two were positive.

**Table 4.** Analysis of variance between the internet and non-internet users in the sense of place.

| Sense of Place [a] | Users | | Non-Users | | F-Value |
|---|---|---|---|---|---|
| | Mean | SD | Mean | SD | |
| Facility satisfaction | 8.24 | 2.24 | 8.98 | 1.54 | 4.527 * |
| Job satisfaction | 7.18 | 2.12 | 6.87 | 1.92 | 0.775 |
| Staying intention | 5.88 | 2..45 | 8.58 | 1.54 | 51.141 ** |

[a] Each variable is the sum of two questions with 5-Likert scale, ** Significant at the 0.01 level, * Significant at the 0.05 level.

Females seemed to be more, but not so much, critical when talking about facilities ($r(187) = 0.182$, sig. (2-tailed) $< 0.05$, 0 for female, 1 for male) because they were generally more often involved in health services as well as more sensitive to lifestyle-related facilities. However, women tended to have intentions for a larger staying intention compared to that of men ($r(187) = -0.201$, sig. (2-tailed) $< 0.01$). For married woman, housing matters were generally left to husband. As for girls, high mobility did not seem to be the preferred thing. Regarding education, residents with higher levels of education tended to have a lower facility satisfaction ($r(187) = -0.430$, sig. (2-tailed) $< 0.01$), especially for those who had formal school in the area with better facilities due to increased self-standards as a logical consequence. It also correlated negatively with intention to stay ($r(187) = -0.487$, sig. (2-tailed) $< 0.01$).

More educated residents felt more daring to stay anywhere if the new place provided a better chance of living. However, as age increased, people's possibility to move out from the village became lower (r(187) = 0.411, sig. (2-tailed) < 0.01).

All these correlations would be dubious in describing the society's condition since the sense of place variables is also affected by demographic background. In other words, this question appeared; is it true that the internet influences the sense of place or it is triggered by demographic background? Therefore, a deeper analysis was conducted to tackle this issue. The point, in this case, is that if there is an obvious difference between users and nonusers on facility satisfaction and staying intention, these results should remain consistent even though the demographic variables are controlled. Otherwise, the relation between tested variables is bias.

As facility satisfaction has a relation with gender and education level, the author compared mean score in the group of females, males, elementary school, junior high school, and senior high school (Table 5). Unfortunately, the analysis of bachelor and above group could not be run because all members were internet users. Between the analyses regarding facility satisfaction, the significant one was only in the female group. Even if the judgment on facility quality was theoretically affected by education level, it did not seem to be significant for the respondents in this study. Hence, the mean difference in facility satisfaction between internet users and nonusers was not consistent.

Since staying intention correlated with gender, age, education, and organization membership, the variance analysis was performed on the group of males, females, elementary school, junior high school, senior high school, village organizer and non-organizer as well as 5 age groups (minus 15–18-aged group due to only 1 nonuser available). The finding revealed that the results were consistent in all groups with several notes. Both in male and female groups, the users had a significantly lower desire to remain in the village. The member of the village organization group did not have a significant difference here. But the author argued that this could not represent the general condition since most of the respondents, 91.4%, were not active as village organizer (see Table 1) and there was a clear difference in this group.

Regarding the educational group, the staying intention notably decreased in line with the educational levels (see Figure 2). However, there was a consistency when comparing the desire to stay in the village between users and nonusers. The mean difference in the elementary school group was significant at the 0.10 level. Members of this group likely tended to have the lower skill to gain in processing the information affecting the degree of difference. Meanwhile, the users and nonusers in a junior high school group had obvious differences in terms of variance analysis in which users' scores were lower than nonusers' score. Consideration needs to be taken when addressing the comparison in senior high school group because there was no noticeable difference. But this result was influenced by the inadequacy of the number of nonusers in the group, i.e., 4, while there were 65 users. Referring to the finding of Ali & Kumar [8], people graduated from senior high school had better skill in obtaining and processing information from the internet to fill limitations. Therefore, the author judged that staying intention difference was consistent in all educational levels.

The variance analysis results in the five age groups regarding staying intention between users and nonusers were all similar with varying significance levels. Not only significant in all groups, the pattern of the mean score of users and nonusers appeared to be unique following the age groups (see Figure 3). Differing from the mean score of users that varied between age groups, the mean score of nonusers remained constant in all age groups. The internet strongly affected the youngest groups; therefore, their staying intentions were very low, indicated by the least average values. The internet is a weapon for young people that are generally at the age of 19–26 and busy building a personal welfare base. However, the users in the older group tended to have a higher degree of desire to stay in the village. The increase in age might cause this. So, the difference of staying intention between users and nonusers was consistent even though the demographic variables were controlled. In other words, internet use decreased the desire to stay in the village but did not affect facility satisfaction and

job satisfaction. Better access to the outside world owned by internet users turned out to reduce the intention to stay in the village.

**Table 5.** Analysis of variance of the sense of place between internet users and non-users based on demographic.

| Sense of Place [a] Variables | Demographic Variables | Demographic Sub-Variables | Users | | Non-Users | | F-Value |
|---|---|---|---|---|---|---|---|
| | | | Mean | SD | Mean | SD | |
| Facility satisfaction | Gender | Male | 8.65 | 2.03 | 9.04 | 1.46 | 0.755 |
| | | Female | 7.43 | 2.42 | 8.92 | 1.63 | 7.654 *** |
| | Education | Elementary sch. | 9.53 | 0.87 | 9.17 | 1.34 | 0.998 |
| | | Junior high sch. | 9.17 | 1.50 | 8.64 | 1.69 | 1.142 |
| | | Senior high sch. | 8.14 | 2.21 | 8.75 | 2.50 | 0.286 |
| | | Bachelor & above | 5.90 | 2.39 | – | – | – |
| Staying intention | Gender | Male | 5.65 | 2.42 | 8.26 | 1.81 | 23.453 *** |
| | | Female | 6.34 | 2.47 | 8.88 | 1.20 | 23.227 *** |
| | Age (years old) | 15–18 | 5.57 | 2.65 | 8.00 | – | – |
| | | 19–22 | 4.93 | 2.24 | 8.50 | 1.38 | 14.329 *** |
| | | 23–26 | 5.67 | 2.27 | 8.75 | 2.50 | 6.051 ** |
| | | 27–30 | 7.39 | 2.08 | 8.57 | 1.60 | 3.291 * |
| | | 31–34 | 7.00 | 2.10 | 8.52 | 1.47 | 7.710 *** |
| | Education | Elementary sch. | 7.82 | 1.70 | 8.73 | 1.55 | 3.474 * |
| | | Junior high sch. | 6.42 | 2.38 | 8.71 | 0.99 | 12.071 *** |
| | | Senior high sch. | 5.31 | 2.52 | 7.00 | 2.45 | 1.698 |
| | | Bachelor & above | 5.19 | 1.86 | – | – | – |
| | Village organizer | Member | 7.00 | 2.16 | 8.67 | 1.51 | 2.734 |
| | | Nonmember | 5.80 | 2.46 | 8.57 | 1.56 | 47.161 *** |

[a] Each variable is the sum of 2 question with 5-likert scale; *** Significant at 0.01 level; ** Significant at 0.05 level; * Significant at 0.10 level.

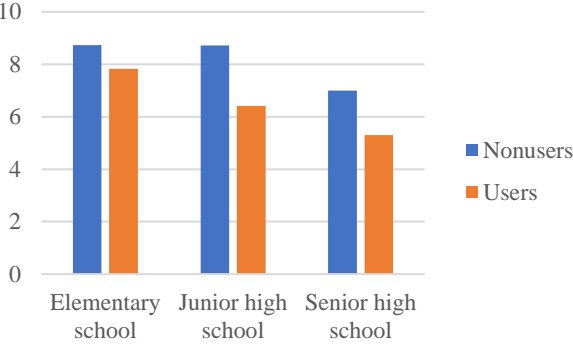

**Figure 2.** Mean comparison of staying intention between internet users and nonusers based on education level.

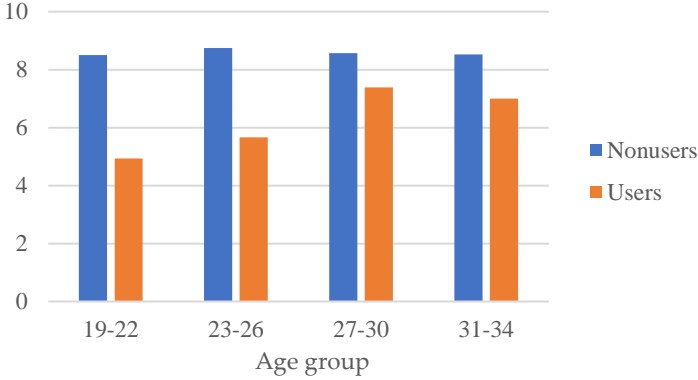

**Figure 3.** Mean comparison of staying intentions between internet users and non-users based on the age group.

3.2.2. The Impacts of the Internet on Local Participation

The weekly event which was attended by more villagers was the dhikr recitation. 58.3% of non-internet users have attended this event (Table 6). Most of them claimed to attend often (33.3%) and always (12.5%). The rests were classified as half (6.3%) and rarely (6.3%). Meanwhile, more than three-fourth of the users had attended dhikr recitation with various intensity ranging from always (23.7%), often (30.9%), half (6.5%), to rarely (13.7%). Then, the users group tended to have notably a portion of respondents attending Koran recitation (37.4%) and sport (36.7%). The nonusers' number attending such events was much lower. According to key informants, it is noticed that even among young people, the portion of people engaged in a workout with surrounding friends was relatively small. Only the youngest group was more likely to have a higher intensity in joining sports. While the Koran readings were generally attended by older groups.

**Table 6.** Respondent's attendance at local events.

| Local Events | Users (%) | | | | | Nonusers (%) | | | | |
|---|---|---|---|---|---|---|---|---|---|---|
| | Never | Rarely | Half | Often | Always | Never | Rarely | Half | Often | Always |
| Dhikr recitation | 25.2 | 13.7 | 6.47 | 30.9 | 23.7 | 41.7 | 6.3 | 6.3 | 33.3 | 12.5 |
| Qur'an recitation | 62.6 | 11.5 | 7.2 | 8.6 | 10.1 | 81.3 | 6.3 | 6.3 | 4.2 | 2.1 |
| Workout | 63.3 | 15.1 | 4.3 | 7.19 | 10.1 | 93.8 | 6.3 | – | – | – |
| Neighborhood meeting | 86.3 | 7.19 | 3.6 | 0.7 | 2.2 | 97.9 | 2.1 | – | – | – |
| Youth meeting | 87.8 | 3.6 | 2.9 | 1.4 | 4.3 | 95.8 | 4.2 | – | – | – |
| Independence Day | 61.9 | 10.1 | 13.7 | 4.3 | 10.1 | 85.4 | 4.2 | 2.1 | – | 8.3 |
| Neighbor's festivity | 15.8 | 16.5 | 11.5 | 38.8 | 17.3 | 4.2 | 14.6 | 14.6 | 45.8 | 20.8 |
| Cleaning village | 74.1 | 5.0 | 10.8 | 5.0 | 5.0 | 87.5 | 2.1 | 8.3 | 2.1 | – |
| Eid al-adha | 64.0 | – | 9.4 | – | 26.6 | 75.0 | – | 4.2 | – | 20.8 |

Among annual events, the most attended activity, both by users and nonusers, was to help in neighbor's festivity. Rural people in Indonesia often conduct such kind of activity. It is a kind of celebration or party related to religion. In the Javanese Islamic tradition, some affairs should be celebrated such as marriage, circumcision, praying for deceased relatives, etc. The villagers seemed motivated to help to neighbor's festivity and the internet nonusers appeared to be involved more in proportion than the users. However, the portion of users was higher in those who had attended other yearly events compared to that of nonusers. Youth villagers seemed not to be motivated to attend neighborhood discussion events. 86.3% and 87.8% of internet users never came to a neighborhood meeting and youth meeting, respectively. Meanwhile, 97.9% and 95.8% of nonusers never attended such affairs. Cleaning the village was also a crucial activity because no other parties would clear the village's common space except the residents. But, in the present research, only 25.9% of internet users and 12.5% of nonusers had joined this activity.

Statistically, the users tended to have higher participation scores at a weekly event compared to nonusers (sig. at 0.01 level) (Table 7). The presence of users also dominated at the yearly events (sig. at 0.05 level). This result was contrary to the authors' hypothesis. Since the users had a better network to the wider space, it was assumed that their physical interaction with the outside world would increase, and their participation in village events would decrease. This result might be caused by the fast communication provided by the internet-based chatting so that the users would easily receive information about village events. Therefore, they had a better chance of attending affairs in the village. However, the bias potential must be detected since demographic characteristics strongly influence participation.

The participation of the villagers in the weekly events was influenced by education (r(187) = 0.198, sig. (2-tailed) 0.01). The higher the education level, the greater the intensity of attendance to the weekly events will be. The participation of villagers in yearly events was also affected by gender, i.e., male was more likely to join these activities (r(187) = 0.167, sig. (2-tailed) < 0.01, 0 for female, 1 for male). Then, people's membership in the village organization encourages their participation both in weekly and yearly events due to two practical reasons: the village leaders could easily mobilize the village

organizer and they also become the figure in organizing the events. Age and monthly spending did not seem to be essential in this case. This correlation, however, could cause bias in the variance analysis regarding the participation between users and nonusers. Hence, a further analysis was necessary.

**Table 7.** Variance analysis of attendance between internet users and non-users on local events.

| Local Participation Variables | Users | | Non-Users | | F-Value |
|---|---|---|---|---|---|
| | Mean | SD | Mean | SD | |
| Weekly events [a] | 6.92 | 3.13 | 5.15 | 2.10 | 13.313 ** |
| Yearly events [b] | 11.59 | 3.89 | 10.29 | 2.70 | 4.575 * |

[a] Sum of three events with 5-attendance scale; [b] Sum of six events with 5-attendance scale; ** Significant at the 0.01 level; * Significant at the 0.05 level.

Since variable of weekly events correlated to education level and membership in village organization ($r(187) = 0.285$, sig. (2-tailed) $< 0.01$), variance analysis was performed in the groups of elementary school, junior high school, senior high school, member, and non-member of organization (see Table 8). There was no clear difference in an elementary school group. It was probably because the internet users in this group did not have a good initiative in networking with surrounding people through the internet, and they only have limited skills to use the internet. But, in junior high school, there was a clear difference between users and nonusers as users had a higher participation score. Those similarities were observed in the staying intention analysis in which there was no obvious difference in senior high school groups due to the limited number of non-users. The users, either the member or the non-member of the village organization, also tended to have higher participation scores.

The deeper analysis of variance was run in the group of male, female, member, and nonmember in checking the consistency of the statistical differences in participation in yearly events between the users and nonusers. The variable of the weekly event was correlated to gender and membership in the village organization. There was no clear difference among genders in the group. Even though the users in the village organizer group had significantly higher participation, it was still far from the consistent result as there was no obvious difference in the nonmember group. Finally, based on all the results, the authors considered that the mean difference in the score of the weekly event was consistent, while that in the yearly events was not. Therefore, it was judged that internet use increases the participation of the people in weekly events but did not have a remarkable effect on the villagers' participation in yearly events.

**Table 8.** Analysis of variance of local participation between internet users and non-users based on demographics.

| Village Events | Demographic Variables | Demographic Sub-Variables | Users | | Non-Users | | F-Value |
|---|---|---|---|---|---|---|---|
| | | | Mean | SD | Mean | SD | |
| Weekly events [a] | Education | Elementary sch. | 6.06 | 3.42 | 5.10 | 2.23 | 1.354 |
| | | Junior high sch. | 6.78 | 2.79 | 5.00 | 2.11 | 4.631 * |
| | | Senior high sch. | 7.32 | 3.33 | 6.00 | 0.82 | 0.621 |
| | | Bachelor & above | 6.62 | 2.82 | – | – | – |
| | Village organizer | Member | 11.40 | 2.37 | 5.67 | 2.34 | 22.201 ** |
| | | Nonmember | 6.57 | 2.91 | 5.07 | 2.09 | 9.542 ** |
| Yearly events [b] | Gender | Male | 11.95 | 3.99 | 10.91 | 2.81 | 1.366 |
| | | Female | 11.25 | 3.55 | 10.48 | 2.80 | 1.646 |
| | Village organizer | Yes | 16.00 | 5.42 | 9.00 | 1.26 | 9.458 ** |
| | | No | 11.25 | 3.55 | 10.48 | 2.80 | 1.646 |

[a] Sum of three events with 5-attendance scale; [b] Sum of six events with 5-attendance scale; ** Significant at the 0.01 level; * Significant at the 0.05 level.

### 3.3. Internet Effect on Social Capital (RQ2)

### 3.3.1. Involvement in the Circles

Respondents' ownership of relationship circles varies according to individual demographic backgrounds. In rural life, being active in supporting activities and interregional organizations is not common. Among them, only 6.5% of internet users and 2.1% of non-internet users (see Figure 4) were involved in the circle of supporting activities i.e. additional group activities in which people can actualize their hobbies or increase skills and knowledge such as non-scholastic courses, skills training, hobby groups, and others. Still in a small portion, respondents involved in interregional organizations, e.g. cultural or political organizations across villages or regions are only 3.6% for internet users and 2.1% for the non-users. We also found that users joining online communities are not so striking, just 3.6%. Different to social media group of school- or work-based interaction, online community here is the gathering of geographical dispersed people with similar interests on the online platform without a history of strong friendships among members such as job hunters, online sellers or scholarship hunters. Some of our respondents are migrants (16.5% for the users and 25.0% for the non-users). They have moved from their original village to Tamansari Village, the place we conducted the research in. Almost all of them said that they moved due to marriage reason. While the portion of people who have relationships based on their activities in the college is only 15.1%. They are all internet users.

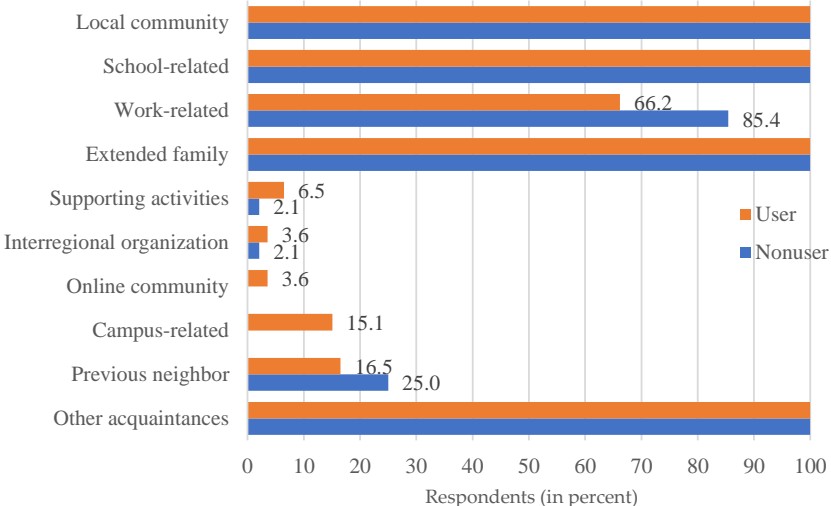

**Figure 4.** Respondents' involvement in the circles of relationship.

Local community, school-related, extended family and other acquaintances are the circle of relationships that all respondents have. These four plus work-related circles are the most common medium in which people get relationships as the first relationships people have are family and local people, then life journey force them to enter school and working world. Share of the working people is notable as well, but jobless, household, and active student groups don't have such relationships from this circle. For further analysis of social capital comparison, we only use these five circles to balance the respondents and to reduce bias possibilities. Local community circle was used to analyze local social capital, while school-related, extended family and other acquaintances are for broad social capital. Working-based relationships have been employed only when addressing working and non-working people only.

### 3.3.2. Comparison of Social Capital

The challenge of this research was how to differentiate one's SC gained from surrounding neighborhood relationships and from broader networks. This was to show that undesired impacts arise along with the wider network owned by internet users. Different from the assumptions of this study,

internet users generally got a better occasion to increase their acquaintances inside the village. As a result, their bridging became higher than that of the nonusers (sig. at 0.05 level; see Table 9). However, the bonding level of the users and nonusers was not significantly different. They tended to have the same quantity of bonding. But with the same level of bonding, internet users had a better quantity of acquaintance that would help them if they asked for certain help (sig. at 0.05 level). Regarding the broad SC, internet users had better quantity in bridging, bonding, and social support – all are significant at 0.01 level.

**Table 9.** Variance analysis regarding local and broad social capital of internet users and non-users.

| Social Capital | Social Capital Dimensions | Users | | Non-Users | | F-Value |
|---|---|---|---|---|---|---|
| | | Mean | SD | Mean | SD | |
| Local social capital | Bridging | 7.54 | 1.58 | 6.94 | 1.24 | 5.761 * |
| | Bonding | 7.12 | 1.58 | 6.98 | 1.25 | 0.325 |
| | Social support | 6.86 | 3.46 | 5.71 | 2.00 | 4.774 * |
| Broad social capital | Bridging | 19.43 | 2.84 | 17.06 | 2.96 | 24.264 ** |
| | Bonding | 17.80 | 3.40 | 15.50 | 2.40 | 16.293 ** |
| | Social support | 16.96 | 7.50 | 11.21 | 4.40 | 25.190 ** |

\*\* Significant at the 0.01 level; * Significant at the 0.05 level.

The impact of internet use on local and broad SC also varied based on the occupation owned by the respondents. In local SC, the non-users in the working people group were only better in bridging quantity (sig. in 0.1 level only; see Table 10). The quantities of local bonding and local social support did not have a significant difference compared to those of the nonusers, even though the mean scores of users were higher. The consequence of people working, especially outside the village, was the lower intensity of interaction with the local community. Personal SC will only be created through enough (face-to-face) interaction between individuals. The interaction increases shared memory, which then strengthens the relation ties. Conversely, the working people using the internet gained better broad SC in all three dimensions than they who do not.

**Table 10.** Variance analysis regarding the social capital of the internet users and non-users in the groups of working and non-working people.

| Occupations | Social Capital | Social Capital Dimensions | Users | | Non-Users | | F-Value |
|---|---|---|---|---|---|---|---|
| | | | Mean | SD | Mean | SD | |
| Working | Local social capital | Bridging | 7.74 | 1.42 | 7.21 | 1.25 | 2.962 * |
| | | Bonding | 7.21 | 1.62 | 7.00 | 1.25 | 0.379 |
| | | Social support | 6.94 | 3.58 | 6.00 | 1.59 | 1.777 |
| | Broad social capital | Bridging | 18.53 | 3.01 | 16.04 | 3.26 | 13.506 *** |
| | | Bonding | 16.65 | 3.26 | 14.50 | 3.70 | 8.319 *** |
| | | Social support | 14.30 | 7.60 | 9.32 | 4.53 | 10.592 *** |
| Non-working | Local social capital | Bridging | 7.72 | 1.06 | 6.63 | 1.12 | 10.860 *** |
| | | Bonding | 7.08 | 1.63 | 7.00 | 1.29 | 0.031 |
| | | Social support | 7.08 | 3.00 | 5.32 | 2.52 | 4.277 ** |
| | Broad social capital | Bridging | 10.84 | 2.41 | 8.95 | 2.09 | 7.440 *** |
| | | Bonding | 8.40 | 2.42 | 8.32 | 2.47 | 0.013 |
| | | Social support | 6.92 | 4.00 | 4.89 | 2.56 | 3.707 * |

\*\*\* Significant at the 0.01 level; ** Significant at the 0.05 level; * Significant at the 0.10 level.

Nonworking people mostly do their activities around their homes so that they had more time to interact with the surrounding residents. The users in this group could also get benefits from the internet since it helped them to get better local bridging (sig. at 0.01 level). Though their bonding remained the same with the nonusers', the users obtained stronger ties with more acquaintances so that more people were willing to help upon his or her request (sig. on 0.05 level). The weakness of people

who only do daily activities around the house is that they are less intensive in managing face-to-face meetings with acquaintances. Internet did help improve their bridging (sig. at 0.01 level), but the formation of strong ties did not occur massively so that there was no real difference between users and non-users of the internet in non-working group. The social support that could potentially be obtained by internet users appears to be slightly more prominent than non-users. Nevertheless, in general, the internet also helps them manage their relationships or networks of acquaintances.

### 3.4. Effect of the Internet Use for Certain Purposes (RQ3)

The users utilized the internet for various purposes. Such utilization could in general be grouped into communication, entertainment, game, self-expression, self-improvement, and self-earning. The authors tried to observe the effect of different purposes in using the internet by villagers toward their social life. Stepwise regression analysis was performed using these purposes as independent variables, and sense of place, local participation, local SC, and broad SC as dependent variables, respectively, in model 1, model 2, model 3, and model 4 (see Table 11). Communication range – how many acquaintances have routine contact through the internet with the user – was also included since it was noticed that intensive communication with less than 10 people was different from that with more than 50 people in producing social ties.

**Table 11.** Internet use for certain purposes in regression with sense of place, local participation, local social capital, and broad social capital.

| Models | | Equations | | | |
|---|---|---|---|---|---|
| | | 1 | 2 | 3 | 4 |
| Dependent variables | | Sense of place | Local participation | Local social capital | Broad social capital |
| Constant | | 24.828 | – | 17.501 | 38.798 |
| | *Intensity* | | | | |
| | Communication | – | – | – | – |
| | Entertainment | – | – | – | – |
| | Game | – | – | – | – |
| Independent variables | Self−expression | – | – | – | – |
| | Self−improvement | −0.412 | – | – | 0.859 |
| | Self−earning | – | – | – | – |
| | *Communication range* | | | | |
| | Local people | – | – | 1.484 | – |
| | Broad acquaintances | −0.287 | – | – | 1.629 |
| R | | 0.313 | – | 0.350 | 0.528 |
| $R^2$ | | 0.098 | – | 0.122 | 0.279 |
| Adjusted $R^2$ | | 0.085 | – | 0.116 | 0.268 |

− excluded (stepwise method).

The regression analysis result showed that one's routine interaction with several local people could lift the local SC since they correlate positively (model 3). Then, there was not any certain internet use affecting the local participation. The intensity of self-improvement was carried out, and the number of broad acquaintances routinely contacting via the internet negatively correlated with the sense of place in the village (model 1). Conversely, they have a positive connection with the broad SC (model 4). Using the internet to improve knowledge or skill and connect more people was good. It increased personal broad SC. But, at the same time, a better understanding of the outside world raises one's preference for higher living standards. If the current living area could not meet such preference, one's dependence on the village would decline. In the sense of place, it was known that staying intention was the most influenced variable.

## 4. Discussion

### 4.1. Internet Impact on Sense of Place and Participation (RQ1)

Our results about the sense of place and local participation in relation to the internet show contradictions but are still reasonable. The striking thing is that the internet on the one hand increases the participation of villagers in local activities, but on the other hand it accelerates the decrease of sense of place in some respects, especially, the most obvious one, the possibility of villagers to move out of the village and live elsewhere. In this case, the results on local participation differ from the initial assumptions we made in this study that the internet decreases the participation of rural youth in local village activities while that on the sense of place is right.

The information network facilitated by the internet helps villagers to mobilize the others to attend neighboring events such as by reminding the location and schedule of events. However, it is not clear whether the distribution of information via the internet is dominant or not in encouraging participation at the local community level because this is not further assessed. Yet, Hampton and Wellman [60] reported that internet users tend to socialize more including in events. people naturally use the internet to find and distribute activities in their communities, as Sproull and Kiesler [61] and Stern et al [11] have shown, besides looking for information about the issues they face. Other studies also show the same thing that the internet is becoming a medium capable of increasing direct human interaction at events in a variety of scopes and issues such as in the economy [62,63], community [56], and politics [64,65], although that happens with different processes.

Then, in the internet era, an increase in local SC (discussed in 4.2) and local participation do not guarantee the attachment of people to their place of origin, not least in the rural area case (see also Onitsuka [66]). This implies that there are other factors that strongly influence one's perspective on the ideal place to live. Stokols and Shumaker [35] argue that if a place is unable to meet one's needs, the sense of dependency will be lower, and the person tends to look for another place considered better. The decision to move from place to place was made in response to the conditions of the people at actual time. Vilhelmson and Thulin [9] argue that the tendency of people to move generally involves a balance between gaining access to better life opportunities everywhere and the need for stability. This balance is influenced by the increase in information provided by the internet, at least by information about standards and lifestyles that brainwash internet users.

The internet inherently encourages people of productive age to look for new jobs, better education, or new housing, and form social contacts in a wider range. In a study of job search activities and labor mobility, Stevenson [67] demonstrated that internet use intensified people's information-seeking activities. This causes them to read more advertisements, register for jobs, and expand their search beyond their labor market area. Because information flows freely and online between regions, it becomes easy to explore employment and education opportunities and comparative places of residence [9]. Dutta-Bergman's [56] claim that internet users tend to stay longer in their original places becomes invalid in the case of rural areas, especially in developing countries, in which limited facilities and sources of income still exist. With people's changing understanding and knowledge about places combined with a larger network of personal contacts via SNSs and Chat applications, routines and online activities can influence the tendency of internet user for making decision on migration [9].

### 4.2. Internet Impact on Social Capital (RQ2)

Dozens of studies have shown that the internet generally increases both quality and quantity of SC and only a small proportion of studies have shown its negative impact (e.g. Nie at al. [68]). Nevertheless, there are still opinions, which the author wants to prove and test, that expanding the network of an internet user can weaken interactions at the local level because it is not possible to increase the number of interpersonal relationships without decreasing the quality of relationships with others [3]. However, the results of this study instead show that the internet in fact increases SC both by expanding network interactions and strengthening local relationships.

The users literally utilize the internet to manage relationships with acquaintances and friends in various circles as humans always want to communicate. They share the witting on each other just through SNSs and chat applications so that distance and time barriers can be reduced. To enhance broad bridging and bonding, the users simply maintain next face-to-face meetings via online communication [18]. This effective and efficient way to conduct social interaction makes internet users prominent than nonusers in terms of both quantity and quality of broad, as well as local, social relationships.

Increased local SC is in line with local participation as it is interrelated and strengthening each other [11]. Authors presume that increased local SC in the village is caused by geographical boundaries, encouraging young villagers to intentionally or unintentionally meet, and the existence of local activities or meetings, scheduled or unscheduled, becomes medium to meet villagers which then conversations happen. Local bridging is formed through this condition with loose effects on local bonding. Hence, local events become crucial tools in maintaining SC in rural areas and the internet helps to mobilize young villagers to it. Moreover, even though users and nonusers have the same bonding level, the first has advantages in managing their bonding. For short, the quality is different. The users run interaction with bonding-typed acquaintances easily and deeper relationships could be formed, reinforcing advantages on social supports. Finally, concerns about declining local SC because of rural youth's intensified interaction with the outside world can be ignored.

Some other important findings in this study also need to be highlighted. Firstly, based on our assessment of circle-grouped relationships and internet usage, there are notable numbers of young villagers who have used the internet to increase their knowledge or know-how and to make money through online platforms. Some internet users have been involved in online communities between regions. This is another sign that the pattern of activities of villagers will change and cannot only be seen from the viewpoint of "village as agricultural land". Second, although the internet generally increases SC, the outcome varies according to occupational background. For example, as shown above, even though some people use the internet, it does not mean that their local SC is better than that of non-internet users.

### 4.3. Effect of Internet Usage for Certain Purposes (RQ3)

People nowadays have used the internet for various purposes. Among these objectives, some affect one's preference for rural area where he lives, i.e. self-improvement and the number of broad acquaintances with whom someone routinely contacts. Both are positively correlated to an increase in wider social networks and negatively to the sense of place to the current living place. Using the internet for self-improvement purposes is not a common skill owned by the villagers. Those who can utilize the internet for this purpose must have required literacy on digital devices, boosting themselves in exploring the internet. Unfortunately, people having this positive ability tend to have high possibility to leave the village. The character of autonomy appears to those who can use the internet to solve their daily problems [3,7].

The intensity of using the internet for communication and entertainment purposes is no longer a determinant for predicting the quantity of social relation. This quantity is specified by the effort to manage communication with a greater number of friends or acquaintances. The more people contacted on regular basis, the more their quantity of social relationship. As has been proofed in this study, routine interaction with large number of local people could increase local SC, while strong broad SC is also determined by regular online interaction with considerable number of broad acquaintances. Despite good for individuals, more communication with broad connections harms one's sense of place in the current living area. Villages with limited facilities are prone to be abandoned by people who have strong interactions with the outside world.

The results indicated that one's wider social relations with parties outside the village have an undesirable impact on a regional perspective. The extent of one's interaction turns out to be negatively correlated with one's desire to remain in the village. The network expansion to other places was

followed by increased knowledge in other areas. Social ties formed with individuals or groups strengthen the existence of individuals elsewhere. From here, one's understanding of comparative places also increases. If the home environment is considered less attractive by individuals, one's tendency to leave the village becomes stronger [31]. The situation leads to the rural areas whose inhabitants interact more frequently with the far-flung groups and less intensive interaction with the surrounding neighborhood.

## 5. Conclusions

This study examines the dystopian view that internet penetration in rural areas can reduce the quality of relations between rural residents at the local community level with a focus on young villagers' sense of place, local participation and social capital. Our findings show that this view is not valid in terms of local participation and social capital. In fact, the internet use can improve their local participation and local social capital. It transformed and supplemented the way of villagers to distribute the information regarding the local events and to maintain the social relationship. But this dystopian view is relevant in the issue of sense of place especially people's possibility to move out from the village. The free-flowing information on the internet encourage young villagers to recognize better life opportunities in other places. The ability in performing self-improvement and maintaining broad social capital through the internet becomes a booster for internet users to reconsider where they will live. This imply that, with the constant increase of internet users in rural area, the young villagers in near future will be more mobile - migration phenomena will be more visible than before.

**Supplementary Materials:** The following are available online at http://www.mdpi.com/2071-1050/12/1/115/s1, File S1: Questionnaire form.

**Author Contributions:** R.A.P. formulated the research design, conducted field survey, collected data, analyzed the data and wrote the paper. K.O., E.R. and S.H. supervised in the process of the study. All authors have read and agreed to the published version of the manuscript.

**Funding:** This research was funded by Graduate School of Agriculture, Kyoto University, and the APC was funded by Graduate School of Agriculture, Kyoto University, and IPB University.

**Acknowledgments:** We would like to thank to local village government for giving permission to conduct the research in the village and to the villagers who participated as respondents, key informants, and assistant during collecting the data.

**Conflicts of Interest:** The authors declare no conflict of interest. The funders had no role in the design of the study; in the collection, analyses, or interpretation of data; in the writing of the manuscript, or in the decision to publish the results.

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
