# Peer review of "Social Interaction of Indonesian Rural Youths in the Internet Age"

_sustainability, doi:10.3390/su12010115_

Round 1
Reviewer 1 Report
Very interesting paper on a timely topic. The authors seem to be struggling with an emerging, though also widely studied, question about lifestyle choices, and the moral and ethical questions raised by the adoption of and adaptation to technology.
The paper seems well-conceived in terms of methodology and statistical analysis, though perhaps a reviewer with more facility in these areas would be better placed to properly evaluate these points. Therefore, I will limit my comments mainly to the apparent, though unstated, assumptions embedded in the language employed.
The literature reviews seems apt, though I would like to see some note of the work of anthropologists who have worked in rural areas. Most of what is cited seems to drawn from studies of urban areas or from within the development discourse.
Several instances of awkward or unclear prose could use smoothing and clarity, e.g. "raises the discourse between scholars and researchers." In another instance, using "also" and "on the other hand" in the same sentence is redundant. There are other cases. A good proofreading by a native speaker would help to improve the prose.
A passage on p. 5, "Place dependence refers to a strong sense of association with the place concerning a goal-oriented and resource-oriented matters in a place that can support [28]. The sense of place linguist into this concept became known later as the two-dimensional model" seems not to make sense, perhaps it's incomplete?
Try to avoid pre-judging phraseology, e.g "the inadequacy of technology restrained the interaction between rural youth and the outside world, while strengthening the interaction within a narrow and limited scope of space," especially in the beginning before presenting findings. Phrase the same idea more neutrally. Using terms like "forced" and "limitation" furthers what could be construed as a preference on the part of the researchers, while the paper itself appears more neutral on the question. For example, the question of "limitations" could be, ad has been understood, in a positive light, especially in the context of foster sustainability, though in this case it is used negatively. With those examples in mind, the paragraph on pp. 1-2 that includes the above passage could be rewritten and more inviting to thought, and the authors could look for other instances of this unintentional bias through word choice. It'd make the paper more inviting.
Similarly, speaking of "better literacy" as on p. 2 indirectly suggests that the villagers are "illiterate," which appears at odds with the vibrancy of oral and communal culture that the paper posits side by side with the arrival of media. It's a little bit of a slippery discourse, as the word choices we use, the metaphors we employ, are imparting information that the authors may not actually be intending.
The phrase on p. 2 about "shopping, playing games, reading, and watching videos" suggest that these activities were absence prior to use of the internet. It may be more accurate to say that these kinds of activities were changed, altered or filtered or modified. This is another example of habitual prose that can be easily improved by being more attentive to how the word choice contradicts or shapes the thought it is supposed be expressing. (These ideas have been laid out by Bowers, who has written on the metaphorical use of language, and Norberg-Hodge, who noticed similarly culture changes with the introduction of technology in Ladakh, India.)
Speaking of accessing "external resources" suggest, again along the lines of what was noted above, that the villages are impoverished, while from the description provided by the authors suggests a lifestyle abundant in community and other resources. I suggest rephrasing the framework toward a more qualitative focus, moving away from the presence or absence of technology, literacy, resources, etc. and more toward a nuance focus on how these are being redefined by technology.
The paper is sending mixed messages about local/global dynamics. On the one hand suggesting the benefits of partaking in impersonal or virtual networks and on the other suggesting that fora and chat can strengthen local networks. This inbuilt contradiction in speaking about technology is not adequately examine in the paper.
Phrases like "Indonesia's rural areas have a unique tradition of social interaction" again seem at odds with the discussions of the benefits of technology. It almost appears that there are two opposing, though unacknowledged (perhaps taken-for-granted?) point of views embedded in the paper. The paper would certainly benefit from mutual reflective discussions among the authors about these issues.
A passage on p. 6, "By contrast, bridging is good for connection to external assets that link somebody to distant acquaintances or move him in a new different circle [15]. Most people get job information as one of the desired outcomes from this weak tie [38]. For short, both strong and weak ties of social capital have their crucial function for individuals," is another example of being vague and awkward. This passage also points to an overuse of "good" and "bad," which pervades the paper to the point of appearing meaningless. The paper could be improved by providing a more qualitatively nuanced discussion of these ethically and morally-laden terms.
Something appears off in a passage on p. 12, "Using the internet for self-improvement and self-learning was considered as an advanced utilization since the users necessarily have better skills to grab this broad internet potential. As the author found in the correlation test, these had positive relation to education level." Use of passive voice is confusing terms of actors and actions, and referring to the author in a jointly authored paper is somewhat perplexing.
On p. 21, in the passage "If a place is unable to meet one's needs, the sense of dependency will be lower, and the person tends to look for another place considered better," the concepts of "needs" is contentious and has been problematized by development theorists, especially if it is understood as "imputed lacks." Some discussion perhaps about the difference between wants, needs and necessities would go a long way to clarifying what the authors appear to be saying in this paper.
On the same page, the passage "Those using the internet tended to have lower desires to stay in the village. The use of the internet gives awareness to the village youth that other places can provide what they do not get from in their residence," seems to be at the crux of what paper is attempting to evaluated, but as noted above the unclear use of metaphor-laden language obfuscates the point.
On p. 23, "Therefore, setting a new modern economic plan for rural areas involving young villagers is crucial," while the point about economic opportunities is central to what the authors are saying, the value-laden term "modern" is another example of how the prose and word choices seem to contradict the author's views.
Author Response
Thank you so much for your review of my article. I have accommodated your comments, but it seems that not all things are well accommodated. I will try to mention in several points.
As for the two-dimensional model term, I chose to delete the term. This term appears among environmental psychology researchers who have tried to include other dimensions (social and environmental) as part of the sense of place. Because the latest construct does not fit into my research, I used the concept of sense of place relating to place dependence, presented by Stokols and Shumaker. The author discusses 4 topics (internet usage, sense of place, local participation, and social capital) which are partially correlated. However, the literature containing internet usage are mostly universal and only a few discuss the rural environment. Finally, the author tried to draw several phenomena, analyzes, and theories that are still relevant to be applied in a rural context. Regarding the results of anthropologist work, unfortunately, anthropologists’ writings discussing Java generally contain political and traditional economic nuances with no one, in our search, addressing cultural and social activities. The writer has difficulty in filtering the thought from available papers for being associated with this one. Finally, throughout the writing, the writer tried to avoid economic or political discussion as it would make the paper more ambiguous. Lecturers’ books about the social life of rural Indonesia are also not easy to find because academic books sold in e-book format are still rare in Indonesia. In the first paragraph of Introduction (1.1) and in Rural Traditional Events (2.4) of the Literature Review, the author attempted to portray rural in traditional view, historically influenced by past conditions. The author tried to direct the explanation to the possibility of disruption in the internet age. We added a few words to emphasize this point instead. The author is not too familiar with reviewer comments relating to "metaphor-laden language". The first draft has been proofread by native speakers. I have also asked him about this but there was no response. We are so sorry about this.Throughout this revision process, the authors realized that the literature review and research framework section still need improvement, but it might take time. Hopefully there is still a subsequent revision process that allows us to improve these parts.
Best Regards,
Ardy Priatama
Rural Planning Laboratory
Graduate School of Agriculture
Kyoto University
Reviewer 2 Report
Authors conducted interesting research regarding the effects of the internet on young people in rural areas of Indonesia with a focus on the sense of place, participation in local activities, and social capital. Paper is very interesting but nonetheless, some parts should be improved such as research background and discussions. Both the research design and the methods are appropriate and adequately described thus being the strongest parts of this manuscript. Although, it is not clear are the findings applicable worldwide or not. The following are some suggestions for improvement.
Suggestions for improvement:
The title is too long. The Authors should consider changing it into somewhat shorter highlighting the importance of the overviewed effects and sustainability goals. The Paper structure is good but there is a significant lack of scientific explanations and discussion. Both research goals and hypotheses are missing. The Authors are urged to highlight and connect them with sustainability issues. Also, in detail discuss the differences between previous researches and this one. A literature review is very broadly written with a lot of irrelevant and/or outdated literature. Only 7% of all used literature are journal articles published during the last five years. Please update your literature review with recent and relevant scientific articles. It is not clear how the proposed model is verified. Refer to the model verification in section 4. Also, add some additional discussion of findings in relation to the research framework as well as research goals and hypotheses are needed. The Authors are urged to draw more specific conclusions.Overall, I strongly urge the Author to reconsider the above-mentioned comments, rewrite the paper accordingly, and resubmit.
Author Response
Thank you so much for your review of my article. I have accommodated your comments, but it seems that not all things are well accommodated. I will try to mention in several points.
In this article, we attempted to examine the dystopian view claiming that the internet can reduce the quality of social relations at the local community level. This issue in rural developing countries is still very rarely discussed. We emphasize on this point in the introduction and we have tried to mention again in certain sections. Then regarding the hypothesis, from the beginning, due to the limited discussion in the rural environment, we brought the flow of writing in the corridor of research questions. These questions were made to test the dystopian view. The challenge faced by the author when writing this article was how to summarize 4 issues at once (internet use, sense of place, local participation, and social capital) in brief and clear explanations. Literature that specifically addresses the rural environment is also quite rare, especially in developing countries. Finally, the author tried to draw several phenomena, analyzes, and theories that are still relevant to be applied in a rural context. As for model verification, I hope to get more detailed suggestion related to this because my supervisor considers that it is enough.Throughout this revision process, the authors realized that the literature review section and research framework still need improvement, but it might take time. Hopefully there is still a subsequent revision process that allows us to improve these parts.
Best Regards,
Ardy Priatama
Rural Planning Laboratory
Graduate School of Agriculture
Kyoto University
Round 2
Reviewer 2 Report
The Authors conducted interesting research regarding the effects of the internet on young people in rural areas of Indonesia with a focus on the sense of place, participation in local activities, and social capital. In revised version authors gave additional insights into their research and also gave all required clarifications. Overall, I believe that the article provides valuable content to the present body-of-knowledge.